# Evaluation of the Bioactive Compounds of *Apis mellifera* Honey Obtained from the Açai (*Euterpe oleracea*) Floral Nectar

**DOI:** 10.3390/molecules29194567

**Published:** 2024-09-25

**Authors:** Sara R. L. Ferreira, Jéssica L. Araújo, Marly S. Franco, Camilla M. M. de Souza, Daniel S. Pereira, Cláudia Q. da Rocha, Hervé L. G. Rogez, Nilton A. Muto

**Affiliations:** 1Center of the Valorization of Amazonian Bioactive Compounds—CVACBA, Federal University of Pará, Belém 66075-750, PA, Brazil; sara.ferreira@icb.ufpa.br (S.R.L.F.); jessica.araujo@icb.ufpa.br (J.L.A.); marly.franco@icb.ufpa.br (M.S.F.); camilla.souza@icb.ufpa.br (C.M.M.d.S.); herverogez@gmail.com (H.L.G.R.); 2Embrapa Eastern Amazon, Belém 66095-903, PA, Brazil; daniel.pereira@embrapa.br; 3Laboratory of Chemistry of Natural Products, Federal University of Maranhão—UFMA, São Luís 65080-805, MA, Brazil; rocha.claudia@ufma.br

**Keywords:** africanized honey bee, antioxidant, bioactive compounds, *Euterpe oleracea*, GC-MS/MS

## Abstract

The biodiversity of Brazil provides an excellent climate and favorable pollination conditions for *Apis mellifera* L., especially in the Eastern Amazon region, which boasts vast floral wealth, including an abundance of açaí (*Euterpe oleracea*) flowers and fruits. In the present study, seven types of honey were evaluated: three containing floral nectar from açaí (Açaí honey) collected in the Eastern Amazon region (Açaí honey from Breu Branco (AH1 and AH2) and Açaí honey from Santa Maria (AH3), both from the state of Pará, Brazil) and four honeys from different regions of Brazil (Aroeira honey from Minas Gerais, Cipó-Uva honey from Distrito Federal, Mangue honey from Pará, and Timbó honey from Rio Grande do Sul). The characteristics of these honeys were evaluated by examining their physicochemical properties, melissopalynological aspects, color, antioxidant potential, and their constituent compounds, which were confirmed through GC-MS analysis. Açaí floral nectar honeys presented physicochemical results similar to those of other honeys, aligning with Brazilian legislation norms, but differed in their high values of free acidity, apparent sugars, and lower reducing sugars, which are directly related to their botanical origin. These differences correlate with unique flavor and distinct aroma characteristics. Melissopalynological analysis confirmed the botanical origin of the honeys containing açaí floral nectar, which had a color range from amber to dark amber. The three açaí honeys demonstrated high antioxidant capacity and superior flavonoid and polyphenol content compared to other honeys, particularly the açaí honey from Breu Branco (AH1), which had four times the content to combat free radicals compared to the honey with the highest potential (Aroeira honey). GC-MS analysis confirmed the presence of antioxidant properties as well as potential anti-inflammatory, antibacterial, antimicrobial, and antitumor capabilities in açaí honeys, which have not yet been fully studied.

## 1. Introduction

Brazilian honey is widely recognized for its high quality on the international market, due to the country’s vast biodiversity, which offers ideal conditions for the production of honeys with distinct characteristics. The combination of the unique climate and botanical diversity contributes to the creation of honeys with unique flavors, aromas and sensory properties [1]. The *Apis mellifera* L., known for its exceptional pollination capabilities, competes effectively with native bees. It plays a key role in the ecological community by interacting with various plant species [2]. According to the Food and Agriculture Organization of the United Nations, Brazil ranks 11th globally in honey production and has significant export potential to high-consumption markets such as Europe and the United States [3,4].

The Eastern Amazon region, particularly the State of Pará, is a major producer of bee products such as honey, propolis, and pollen. This region accounts for 59% of the honey production in northern Brazil [5]. Amazonian honey is distinguished by its unique sensory properties, including flavor, acidity, and aromatic notes, in comparison to honey from other regions produced by either native stingless bees or *Apis mellifera*. Honey also possesses antimicrobial, antioxidant, and anti-inflammatory properties that help prevent oxidative stress, aging, and chronic diseases, mainly due to its high flavonoid and phenolic content [6]. Previous studies indicate that darker honey generally contains more phenolic compounds and exhibits better antioxidant capacities than lighter honey [7,8,9,10].

The introduction of bees to pollinate açaí groves has shown significant results, with a productive increase of up to 30% in crops. According to Muto et al., stingless bees (*Scaptotrigona aff. postica*) have stood out in this process, not only improving pollination but also enabling the recording of monofloral açaí honey [11]. This honey is sourced from the nectar produced by both the male and female flowers of the açaí palm tree (*Euterpe oleracea*). Furthermore, beekeepers have inserted Africanized bees (*Apis mellifera* L.) into açaí groves, resulting in honey that is considered a delicacy in the region due to its unique characteristics. This honey, with its distinct particularities, reflects the diversity of the local flora and the specific interaction between bees and açaí nectar [12].

Açaí honey, produced in the Amazon, stands out due to its unique bioactive compounds derived from the açaí plant. These compounds, which are present not only in the fruit but also in various parts of the plant, such as the floral nectar, include anthocyanin and non-anthocyanin flavonoids, amino acids, fatty acids, minerals, and small amounts of monoterpenoids and nor-isoprenoids. These bioactive compounds are known for their antimicrobial, anti-inflammatory, antiproliferative, gastroprotective, and antinociceptive properties [13]. The combination of these properties with those of honey potentially enhances its health benefits, adding aggregate value to this genuine Amazonian product.

This study aims to characterize açaí honey produced by Africanized bees (*Apis mellifera*) from different regions and to compare the bioactive compound profiles of açaí floral nectar honey with those of wild floral nectar honey from various Brazilian areas, using physical-chemical parameters, quantification of phenolic compounds through spectroscopic methods, and chemical profiling by gas chromatography coupled to mass spectrometry (GC-MS).

## 2. Results

### 2.1. Color and Color Intensity

The results obtained in the analysis were measured in triplicate and expressed by calculating the mm Pfund, using the average analyzed on the Pfund scale (Table 1). Intensity was calculated based on the difference in absorbance read at 450 and 720 nm on the spectrophotometer. 

The honeys with açaí floral nectar present in their composition were in the amber to dark amber range. The Açaí honey (AH1) showed results corresponding to 380.0 mm in the dark amber color range with an intensity of 2.361 mAU (B). In Açaí honey (AH2), the color results were 72.2 mm in the amber range and the intensity was 0.953 mAU (C). Açai honey (AH3) presented a color of 90.3 mm in the amber range with an intensity of 1.463 mAU (Figure 1A).

The honeys of polyfloral origin varied from dark amber to light amber on the mm Pfund scale. Aroeira honey presented a color of 223.3 mm, in the dark amber color range, with an intensity of 1.093 mAU (D). Mangue honey obtained 61.7 mm on the Pfund scale, in the light amber range, with a low intensity of 0.244 mAU (E). Cipó-Uva honey presented 42.2 mm on the Pfund scale, in the extra-light amber range, with an intensity of 0.111 mAU (F). Timbó honey presented a light amber color with 72.0 mm Pfund and intensity of 0.208 mAU (G).

### 2.2. The Melissopalynological Analysis 

The results of the melissopalynological assay indicate a great significance to its botanic and geographical origin [14]. Moreover, the pollen identification results were based on the palinoteca of the Center for Bioactive Compounds (CVACBA).

For all the honeys, a count of 500 pollen grains was conducted, identifying various types of pollen. The AH1 honey contains 50.5% *Euterpe oleracea* pollen, 20.48% *Sida galheirensis* pollen, 25.25% *Acacia sp*. pollen, and 3.78% *Euphorbiaceae* pollen. The AH2 honey was found to contain 44.98% *Sida galheirensis* pollen, 20.8% *Euterpe oleracea* pollen, 17.8% *Euphorbiaceae pollen*, 3.42% *Acacia sp.* pollen, and 3% of other pollen. The AH3 honey consists of 45.8% *Euterpe oleracea* pollen, 26.33% *Sida galheirensis* pollen, 17.33% *Euphorbiaceae pollen,* 7.87% *Acacia sp*. pollen, and 2.67% of other pollen (Figure 2).

### 2.3. The Physicochemical Analysis

The results of the physicochemical analyses were expressed as means and standard deviations and compared to the values required by Normative Instruction No. 11 of the Ministry of Agriculture and Supply [15], as shown in Table 2.

#### 2.3.1. Free Acidity Content

Among the commercial honeys, Aroeira honey had the highest acidity content, with an average of 30.74 ± 0.5 mEq/kg. Cipó-Uva honey had acidity values of 14.62 ± 0.0 mEq/kg, Mangue honey 12.31 ± 0.2 mEq/kg, and Timbó honey 9.19 ± 0.0 mEq/kg. The honeys made with açaí flower nectar from the Santa Maria and Breu Branco regions (AH3, AH1 and AH2) had higher (*p* < 0.05) acidity values, 75.97 ± 0.8 mEq/kg, 73.60 ± 2.9 mEq/kg and 63.03 ± 0.2 mEq/kg, respectively.

#### 2.3.2. pH Values

Although pH analysis is not mandatory for the quality control of Brazilian honey, legislation stipulates that the pH of *Apis mellifera* honeys must be between 3.3 and 4.6 for human consumption. The pH is crucial for assessing the quality of honey, as it influences its texture, stability, and shelf life. Deviations in pH values can indicate adulteration or fermentation of the honey. Lower pH values affect sensory characteristics, while higher values reduce microbial control [16,17].

Analyzing the pH values obtained, honeys from açaí flower nectar showed more acidic pH levels compared to commercial honeys: Cipó-Uva honey had a pH of 3.87 ± 0.01, Aroeira honey 4.70 ± 0.1, Mangue honey 4.20 ± 0.0, and Timbó honey 4.23 ± 0.02, which corroborates the free acidity results. Honeys containing açaí flower nectar had pH values of 3.44 ± 0.00 for Açaí honey AH3, 3.54 ± 1.7 for AH2 and 3.35 ± 0.0 for AH1. 

#### 2.3.3. Honey Moisture Content

Moisture is a critical parameter for honey storage, with Brazilian legislation requiring a maximum content of 20 g/100 g. It can be influenced by the bee species, floral origin, harvest time, and climatic conditions, and it also affects properties such as color, viscosity, and flavor [15,18,19]. All the analyzed honey samples comply with Brazilian legislation regarding moisture content (*p* < 0.05), except for AH2 honey. The higher moisture content in AH2 (*p* < 0.05) may be attributed to its botanical origin as a bifloral honey, whereas AH1 and AH3 are monofloral. Honeys from açaí flower nectar showed moisture content of 19.82 ± 0.0 for AH1, 20.70 ± 0.0 for AH2, and 19.50 ± 0.0 for AH3, while commercial honeys had values of 16.50 ± 0.0 for Cipó-Uva and Aroeira, 19.40 ± 0.0 for Mangue, and 17.30 ± 0.0 for Timbó.

#### 2.3.4. Soluble Solids Values (°Brix)

°Brix analysis quantifies soluble solids in a sample. While Brazilian legislation does not require °Brix analysis for honey quality control [15], it was conducted here for comparative purposes. Soluble solids levels in açaí floral nectar honeys AH1, AH2, and AH3 were 78.50 ± 0.0 g/100 g, 77.67 ± 0.0 g/100 g, and 78.80 ± 0.1 g/100 g, respectively. Commercial honeys had soluble solids levels of 81.77 ± 0.0 g/100 g for Cipó-Uva, 81.77 ± 0.2 g/100 g for Aroeira, 79.00 ± 0.0 g/100 g for Mangue, and 81.03 ± 0.0 g/100 g for Timbó.

#### 2.3.5. Reducing Sugars Content

Reducing sugar content serves as an indicator for distinguishing floral honey, honeydew honey, and their mixtures. Brazilian legislation mandates a minimum of 65% for floral honey and 60% for honeydew honey and its mixture with floral honey [15]. Reducing sugar values below 65% may indicate immature honey [20]. Aroeira honey had a reducing sugar content of 62.26 ± 0.5 g/100 g, while honeys containing açaí pollen AH1, AH2, and AH3 showed values of 62.76 ± 0.2 g/100 g, 64.26 ± 0.6 g/100 g, and 64.49 ± 0.4 g/100 g, respectively. The highest levels of reducing sugars were found in Mangue (71.08 ± 0.3 g/100 g), Timbó (68.06 ± 0.4 g/100 g), and Cipó-Uva (67.75 ± 1.1 g/100 g) honeys.

#### 2.3.6. Apparent Sucrose Content

The high sucrose content may be related to the botanical origin and early harvesting of the honey, since the sucrose has not been completely converted into glucose and fructose by the action of the invertase enzyme [20,21]. The honeys analyzed containing pollen from açaí flowers had an apparent sucrose content of 4.59 ± 0.0 g/100 g for AH3, 5.57 ± 0.3 g/100 g for AH1, and 6.21 ± 0.3 g/100 g for AH2, while commercial honeys showed 3.53 ± 0.1 g/100 g for Cipó-Uva, 5.60 ± 0.1 g/100 g for Aroeira honey, 0.74 ± 0.0 g/100 g for Mangue honey, and 1.38 ± 0.0 g/100 g for Timbó honey. 

### 2.4. Antioxidant Activity and Determination of Bioactive Compounds

The results of the spectrophotometric analyses (polyphenol content, flavonoids content, flavonol content and DPPH) can be viewed as means and standard deviation in Table 3.

#### 2.4.1. Total Polyphenols Content 

The analysis of the total polyphenols was expressed in mg equivalents of gallic acid per 100 g. The results for açaí honeys AH1, AH2, and AH3 were 291.8 ± 8.1, 79.49 ± 2.4, and 118.20 ± 5.2, respectively. For the other honeys, the results were: Aroeira honey, 100.05 ± 0.8; Timbó honey, 34.39 ± 2.3; Mangue honey, 32.24 ± 0.8; and Cipó-Uva honey, 25.15 ± 0.7. The honey produced from the floral nectar of açaí showed statistically significant differences compared to wild honeys (*p* < 0.001). However, only the AH2 honey did not show a statistical difference compared to Aroeira honey. The results can be seen in Table 3).

#### 2.4.2. Total Flavonoid Content 

The analysis of total flavonoids was expressed in mg equivalents of rutin per 100 g. The açaí honeys yielded results of 106.05 ± 5.3 for AH1, 14.36 ± 0.9 for AH3, and 9.71 ± 0.3 for AH2. For the other honeys, the results were: Aroeira honey, 30.24 ± 0.9; Timbó honey, 9.67 ± 0.2; Mangue honey, 7.43 ± 0.6; and Cipó-Uva honey, 3.69 ± 0.1. The AH1 honey showed a statistical difference compared to the wild honeys (*p* < 0.001), while the AH3 honey showed a statistical difference only compared to the Mangue and Cipó-Uva honeys (*p* < 0.05). The results can be seen in Table 3.

#### 2.4.3. Total Flavanol Content 

In the analysis of total flavanols, the results were expressed in mg equivalents of catechin per 100 g. The açaí honeys yielded results of 1.95 ± 0.1 for AH1, 1.27 ± 0.1 for AH3, and 1.21 ± 0.1 for AH2. Timbó, Mangue, and Cipó-Uva honeys fell below the calibration curve limit and could not be quantified, except for Timbó honey, which quantified 4.20 ± 0.1. The Aroeira honey showed the highest flavanol content compared to açaí floral nectar honeys, achieving a significant difference (*p* < 0.001). The results are shown in Table 3.

#### 2.4.4. 1,1-Difenil-2-picrilhidrazil (DPPH) 

The DPPH analysis results were quantified in µmol of Trolox equivalents per 100 mL. Açaí honeys exhibited the highest antioxidant activity: AH1 honey at 396.55 ± 10.4, AH3 honey at 157.81 ± 6.9, and AH2 honey at 114.13 ± 3.2. Among the other honeys, Aroeira honey yielded 98.58 ± 1.2, Timbó honey 14.07 ± 0.7, Mangue honey 15.05 ± 0.6, and Cipó-Uva honey 23.06 ± 0.9.

The antioxidant analysis measured by the DPPH method showed that AH1 honey exhibited the highest antioxidant activity (*p* < 0.001), followed by AH3 and AH2, with values ranging from 396.55 ± 10.4 to 114.13 ± 3.2 µmol eq./100 mL. These values are significantly higher than those observed in the other honeys (Aroeira, Timbó, Mangue, and Cipó-Uva), which showed antioxidant activities ranging from 98.58 ± 1.2 to 14.07 ± 0.7 µmol eq./100 mL. This suggests that honeys predominantly composed of açaí pollen have a stronger antioxidant capacity, likely due to higher concentrations of polyphenols and other bioactive compounds. The results are shown in Table 3.

### 2.5. GC-MS Analyses

The chemical characterization, identified by molecule fragmentation pattern compared with mass spectral data, revealed the presence of major compounds considered to be above 3% area.

Four honeys (Timbó, Mangue, Cipó-Uva, and Aroeira) and three honeys with floral nectar (AH1, AH2, AH3) from *Euterpe oleracea* from different regions of Brazil were analyzed to quantify their major compounds. The floral nectar honeys from *Euterpe oleracea* obtained from the state of Pará were analyzed by GC-MS for quantification of their main compounds.

The açaí floral nectar honey from the Breu Branco (AH1) region contained batilol (18.94%), 5-Hydroxymethylfurfural (17.52%), n-Hexadecanoic acid (10.99%), oleic acid (8.39%), and batilol (5.78%) (Figure 3, Table 4).

Açaí honey from the Breu Branco region (AH2), containing *Euterpe Oleracea* pollen, exhibited compounds predominantly present at concentrations above 3%, including lupeol acetate (14.67%), 1-heptacosanol (7.02%), beta-Amyrin acetate (4.63%), 2-methylhexacosane-n-hexacosane (4.35%), 2-methylhexacosane (3.96%), Dotriacontane (3.56%), and n-tetracosanol (3.02%) (Appendix A).

Açaí honey from Santa Maria (AH3) revealed significant compounds such as ethyl oleate (17.45%), batilol (8.91%), heneicosane (6.54%), 2-Methylhexacosane (4.49%), Methyl 9-octadecenoate (4.31%), oleic acid (4.1%), and 5-Hydroxymethylfurfural (4.05%) (Appendix A).

Timbó honey, from Rio Grande do Sul, contained predominant compounds such as 1-hexacosanol (12.42%), oleic acid (12.05%), pentacosane (8.56%), tetratetracontane (7.48%), n-hexadecanoic acid (7.06%), carbonic acid, batilol (6.17%), heneicosane (5.05%), heneicosane (4.49%), tetracontane (4.45%), and 17-pentatriacontene (4.35%) (Appendix A).

Mangue honey, from Pará, featured predominant concentrations of pentacosane (10.47%), tetratetracontane (9.34%), 1-hexacosanol (8.79%), sitosterol (7.49%), Spiro[bicyclo[3.1.1]heptane-2,2′-oxirane] (7.44%), heneicosane (5.98%), and heneicosane (4.06%), 1-Iodotriacontane (3.45%), glyceraldehyde (3.25%), behenic alcohol (3.24%), and oleic acid (3.04%) (Appendix A).

Honey from Distrito Federal (Cipó-Uva) predominantly contains linoleic acid ethyl ester (39.19%), Ethyl-9-octadecenoate (37.99%), octadecanoic acid ethyl ester (10.15%) and ethyl 9-octadecenoate (3.33%) (Appendix A).

Aroeira honey from Belo Horizonte predominantly contains nonadecyl trifluoroacetate (58.05%), 1-chloro-9-octadecene (51.81%), 1-Hexacosanol (51.50%), 13-Docosen-1-ol (50.99%), tetracontane (47.80%), 4-Methoxyphenyl acetic acid (15.45%), tetratetracontane (9.87%), heneicosane (6.03%), oleic acid (4.74%), tetratetracontane (4.21%), tetratetracontane (4.20%), 1-Heptacosanol (3.98%), Eicosane (3.89%), and Oxirane [(hexadecyloxy)methyl] (3.26%) (Appendix A).

The analyzed honeys exhibited constituent compounds known for their various beneficial properties, including anti-inflammatory, antitumor, antimicrobial, antibacterial, antifungal, and antioxidant properties, as previously reported in other studies (Table 5). Notably, honeys predominantly composed of *Euterpe oleracea* pollen demonstrated potent antioxidant capabilities, as evidenced by the results obtained for AH1, AH2, and AH3.

## 3. Discussion

Açaí honeys (AH1, AH3) are characterized as monofloral, each exhibiting distinct sensory aspects, chemical compositions, and microscopic structures, predominantly comprising more than 45% *Euterpe oleracea* pollen. In contrast, Açaí honey (AH2) is classified as bifloral, containing dominant pollen constituents of *Sida galheirensis* and *Euterpe oleracea*. This variation in floral origin within the same region is attributed to the timing of *A. mellifera* bee pollination during *E. oleracea* blooming periods.

According to Kayani et al. (2019), identifying pollen types in honey is crucial as it impacts various properties influenced by the plant species consumed by bees, thereby affecting honey quality [28]. Maieves et al. (2020) categorized pollen presence in honey based on dominance: ≥45% as dominant pollen, 15–44% as companion pollen, 4–14% as rare pollen, and ≤3% as sporadic pollen. The diversity of pollen observed in honey reflects foraging strategies aimed at minimizing competition with other bee species [45].

The predominance of açaí (*E. oleracea*) pollen in honey from Africanized bees (*A. mellifera*) allows beekeepers to differentiate honey based on its floral origin. This distinction not only promotes the development of a sustainable product originating from the Amazon but also serves as a value-added tool for generating new income streams for açaí producers, given that 90.79% of Brazilian açaí cultivation is concentrated in the state of Pará (Viana et al., 2021) [46].

Studies by Puścion-Jakubik et al. (2022) have established a correlation between honey color and intensity with phenolic compound content, with darker and more intense colors indicating higher potential for phenolic content [47]. As depicted in Figure 2, honeys containing açaí pollen (AH1, AH2, AH3—A, B, C respectively) exhibit darker hues compared to commercial honeys (Aroeira, Mangue, Cipó-Uva, Timbó—D, E, F, G respectively).

In addition, the acidity values of the Cipó-Uva, Aroeira, Mangue and Timbó honeys analyzed were within the quality standards for honey established in Brazil (*p* < 0.05), which set a maximum limit of 50 mEq/kg of acidity [15]. A study by Viciniescki, Cordeiro, and Oliveira on honey samples from Rio Grande do Sul reported acidity values ranging from 27.54 mEq/kg to 42.24 mEq/kg [17], similar to our findings. However, açaí honeys exhibited significantly higher acidity (*p* < 0.05), surpassing regulatory limits. Although Aroeira honey showed acidity levels closer (*p* < 0.05) to those açaí flower nectar honeys, significant differences still exist. Variations in honey acidity may be attributed to factors such as nectar sources, glucose oxidase activity, bactericidal action during ripening, and mineral composition, all influencing honey acidity, flavor, and preservation [17,18]. Higher acidity levels indicate potential antimicrobial properties due to the fermentation of sugars and organic acids [21,48].

On the other hand, previous studies have reported honey pH values ranging from 3.0 to 4.50 in samples of *A. mellifera* honey from various floral sources, which are consistent with the values obtained (*p* < 0.05) and those recommended by Brazilian legislation for honey intended for consumption, although pH analysis is not a quality parameter required by the Ministry of Agriculture and Food Supply [15,17,49]. The pH values observed in the açaí floral nectar honeys were within the pH 3.0 range (*p* < 0.05), indicating high-quality products, as pH values below 4 are associated with greater stability and lower susceptibility to harmful microorganisms [50,51].

Brazilian legislation mandates a minimum reducing sugars content of 65 g/100 g for honey [15]. Açaí pollen-containing honeys (AH1, AH2, AH3) and Aroeira honey demonstrated reducing sugar values below this threshold (*p* < 0.05). A study on six samples of *Apis mellifera* honey reported reducing sugar averages ranging from 17.56 ± 3.44 to 83.38 ± 9.63 [21], consistent with our findings (Table 2).

In addition, current legislation establishes a maximum limit of 6 g/100 g of non-reducing sugars in honey [15]. A study evaluating 29 honey samples from Rio Grande do Sul, Brazil, showed apparent sucrose levels ranging from 0.45% to 3.33%, with an average of 1.29% [20], results similar to those of the honeys analyzed (Table 2). However, AH2 honey exceeds (*p* < 0.05) the apparent sucrose limit established by Brazilian legislation, while the other honeys analyzed are within the limit allowed for non-reducing sugars (*p* < 0.05).

Polyphenols, recognized for their antioxidant capacity, are extensively studied for their potential health benefits [52,53]. In total polyphenol analysis, monofloral açaí honeys AH1 and AH3 exhibit rich compositions, highlighting their nutritional and medicinal value in protecting against chronic diseases [54,55]. AH1 honey, with the highest polyphenol content (*p* < 0.001), offers significant antioxidant potential, while AH3, with a lower yet substantial concentration, also represents a health-promoting choice. Variations in polyphenol levels between the two can be attributed to geographical differences [56,57], underscoring the influence of açaí floral nectar on polyphenol content compared to significantly lower values found in Mangue, Timbó, and Cipó-Uva honeys.

Despite being bifloral, AH2 honey containing 20.8% açaí pollen exhibits significant polyphenol content (*p* < 0.001) compared to wild honeys from Timbó, Mangue, and Cipó-Uva. While lower than other açaí nectar honeys, AH2 polyphenol content surpasses that of wildflower honeys. Aroeira honey, known for its dark amber color and high antioxidant capacity, displays higher average phenolic content compared to Brazilian and global honeys [58,59]. However, AH1 and AH3 monofloral açaí honeys show superior polyphenol levels compared to Aroeira honey.

Flavonoids, a subclass of polyphenols with antioxidant and anti-inflammatory properties [60,61], vary in honey composition based on environmental factors and source (pollen, nectar, or propolis) [62,63]. AH1 honey exhibits the highest flavonoid concentration (*p* < 0.001), attributed to açaí nectar as the primary floral source, which accumulates phenolic compounds during development [64]. AH3 honey, also from açaí nectar, maintains a relatively high profile of bioactive compounds, albeit with lower flavonoid levels compared to AH1.

Comparison with AH2 honey and Aroeira, Timbó, Mangue, and Cipó-Uva honeys underscores AH1 superior flavonoid content. AH2 honey, while showing significantly lower flavonoid values compared to AH1, still contains higher amounts relative to other honeys, indicating the influence of açaí floral nectar on final flavonoid content. Aroeira honey, like its total polyphenol analysis, excels in total flavonoid content, demonstrating its high concentration of bioactive compounds among Brazilian and international honeys [65].

Flavanols, another important polyphenol group with antioxidant activity [66,67], are notably abundant in Aroeira honey. High flavanol levels reaffirm the presence of bioactive compounds in Aroeira honey, supported by previous studies [58,59,65]. Except for Aroeira honey, only honeys derived from açaí floral nectar show flavanol presence, highlighting açaí floral nectar’s impact on honey quality and increased antioxidant compounds, although not as predominant as in açaí honey (AH2).

The significant disparity in DPPH antioxidant activity among groups underscores pollen type’s importance in honey composition. Açaí trees, renowned for their high bioactive compound concentration [48,68], reflect in analyzed honeys. Honeys with greater antioxidant activity enhance health benefits, such as oxidative stress protection [69,70] and anti-inflammatory effects [71].

The findings underscore the importance of considering pollen botanical origin when assessing honey’s antioxidant quality. Honeys predominated by açaí pollen not only contain higher antioxidant compound levels but also demonstrate superior antioxidant capacity, enhancing their potential as functional foods promoting health benefits.

The honeys analyzed demonstrated constituent compounds with properties with various actions, such as anti-inflammatory, antitumor, antimicrobial, antibacterial, antifungal, and antioxidant properties previously reported in extract in other studies, as shown in Table 5.

However, *Euterpe oleracea* floral nectar honeys stood out in their antioxidant properties according to the results obtained. These findings collectively confirm the potent antioxidant capabilities of açaí honeys AH1, AH2, and AH3.

In AH1 honey, the antioxidant compound is Heptacosanol of its composition, indicating its high antioxidant capacity (Figure 3 and Table 4). This evidence is supported by the results of DPPH polyphenolic analysis reported in this study, showing that this honey has three times greater antioxidant capacity compared to other honeys.

Thus, these compounds present in honey play a key role in contributing to the health benefits associated with consuming foods rich in polyphenols and antioxidant substances. They belong to the classes of sesquiterpenes and triterpenoids, respectively, known for their diverse biological activities and significant antioxidant capacities due to their chemical structures, which enable them to neutralize free radicals and mitigate oxidative stress, decisive for maintaining cellular health. 

Compounds present in honey, such as sesquiterpenes and triterpenoids, play important roles in contributing to health benefits associated with polyphenol-rich foods and antioxidant substances. The second compound in AH2 is 1-Heptacosanol, which demonstrates potential in reducing oxidative stress [27]. Ethyl Oleate in monofloral AH3 acts as a foraging pheromone with highlighted antioxidant potential [41,42]. 

This evidence confirms that açaí honeys promote antioxidant capabilities and contain bioactive compounds present in their composition with a higher content than other honeys. They also present compounds with antibacterial, anti-inflammatory, antimicrobial, and antitumor activities with great potential for exploration in future studies. 

## 4. Materials and Methods

### 4.1. Honey Samples

Honey samples from the *Apis mellifera* bee were received at the Amazon Bioactive Compounds Valorization Center (CVACBA) (Belém, Brazil). Three of these samples were derived from the nectar of açaí flowers: two from the municipality of Breu Branco (PA, Brazil) (AH1 and AH2) and one from Santa Maria (PA, Brazil). Additionally, samples of commercial honey were obtained from several regions, including Brasília (DF, Brazil), Belo Horizonte (MG, Brazil), Mangue (PA, Brazil), and Timbó (SC, Brazil).

### 4.2. Color Analysis

Color intensity was measured using a spectrophotometric method, which utilizes the difference between absorbances at 450 nm and 720 nm as a parameter for intensity (Beretta et al., 2005; Pascual-Maté et al., 2018) [72,73]. The calculation formula used is: mm Pfund = −38.70 ± 371.39 × abs

### 4.3. Melissopalynological Analysis

The quantitative analysis of pollen grains helps establishing the proportion contributed by each nectariferous plant to the honey, determining its botanical species, geographic origin, botanical origin, and collection time. This analysis aids in identifying honey of unknown or doubtful origin. Standard methodology was used according to Louveaux et al. (1978), followed by the method without acetolysis for pollen sediments, as per Erdtman (1960), with subsequent slide mounting [14,74].

### 4.4. Physicochemical Analysis

The physicochemical characteristics of honey were assessed according to Brazilian legislation [15] to determine its quality and identity criteria. The analysis included reducing sugars, apparent sucrose, free acidity, pH, moisture, and °Brix. To ensure accuracy, all analyses were performed in triplicate.

#### 4.4.1. Determination of Free Acidity

Free acidity was determined using AOAC method no. 962.19 (1998) and the Analytical Standards of the Adolf Lutz Institute (2008) [75,76]. This procedure involved diluting a honey sample and then titrating it with a 0.05 N NaOH solution until reaching the equivalence point at pH 8.5. The free acidity was quantified based on the volume of NaOH solution required to neutralize the acidic components in the honey.

#### 4.4.2. Determination of pH

The pH of the honey samples was measured after diluting 10 g of the sample with distilled water, performed in triplicate. The measurement was conducted using a digital pH meter from Lucadema (model LUCA-210), following the Analytical Standards of the Adolf Lutz Institute (2008) [76].

#### 4.4.3. Soluble Solids (°Brix)

The content of soluble solids was determined by directly measuring the samples using a Hanna Instruments benchtop digital refractometer (model HI96800, Belgrade, Serbia). The measurements were conducted at an ambient temperature of 25 °C to ensure accurate results.

#### 4.4.4. Moisture Determination

The moisture content of the honey was determined according to AOAC method no. 969.38b (1998) and the Analytical Standards of the Adolf Lutz Institute (2008) [75,76]. The procedure involves measuring the refractive index of the honey at a temperature of 20 °C using a Hanna Instruments benchtop digital refractometer (model HI96800). This refractive index is then converted to a percentage of moisture using the Chataway reference table, which has been revised by Wedmore to ensure accuracy in the determination of moisture content.

#### 4.4.5. Reducing Sugars

Reducing sugars were determined according to the AOAC method (1998) and the Analytical Standards of the Adolf Lutz Institute (2008), following the Lane and Eynon procedure [75,76]. This method involves titrating the honey solution with Fehling’s solution, which is reduced at boiling point. Methylene blue is used as the reaction indicator, allowing for the detection of the endpoint of the titration. The amount of reducing sugars is then calculated based on the volume of Fehling’s solution required to complete the reaction.

#### 4.4.6. Apparent Sucrose

Apparent sucrose (non-reducing sugars) was determined according to the AOAC method (1998), using the Lane and Eynon procedure following inversion through acid hydrolysis with hydrochloric acid. After hydrolysis, the solution was neutralized with 20% sodium hydroxide and then titrated with Fehling’s solution to the endpoint. The amount of apparent sucrose was calculated based on the difference between the total reducing sugars before and after hydrolysis, in accordance with the Analytical Standards of the Adolf Lutz Institute (2008) [75,76].

### 4.5. Determination of Total Phenolic Compounds and Antioxidant Capacity

#### 4.5.1. Total Polyphenols Content 

The total polyphenol content was determined using the Folin-Ciocalteu method [77]. Honey samples were diluted in water, vortexed at 2800 rpm for 30 s, and then centrifuged for 20 min at 4 °C and 8000 rpm. The reaction mixture included 500 µL of diluted sample, 250 µL of Folin-Ciocalteu 1N solution, and 1250 µL of sodium carbonate 75 g/L solution incubated for 30 min at room temperature in the dark. Absorbance was measured at 750 nm using a spectrophotometer Spectro Vision, model T80± (PG Instruments Limited, Woodway lane, Alma Park, UK). Results were expressed as mg equivalent gallic acid per 100 g of honey (mg eq./100 g).

#### 4.5.2. Total Flavonoid Content 

The total flavonoid content was determined by the aluminum chloride assay [78]. Similar to the Folin-Ciocalteu method, honey samples were prepared and reacted with 1 mL of aluminum chloride 2% solution for 10 min. Absorbance was measured at 430 nm on the same spectrophotometer. Results were expressed as mg equivalent rutin per 100 g of honey (mg eq./100 g).

#### 4.5.3. Total Flavanol Content 

Total flavanol content was determined using the *p*-dimethylaminocinnamaldehyde (DMACA) method [79]. After dilution and centrifugation of honey samples, the reaction involved 2400 µL of DMACA solution (0.5 mg/mL) and 400 µL of diluted sample for 10 min. Absorbance was measured at 640 nm. Results were expressed as mg equivalent catechins per 100 g of honey (mg eq./100 g).

#### 4.5.4. DPPH Assay

Antioxidant activity was evaluated using the DPPH assay [80]. Honey samples were prepared similarly to the Folin-Ciocalteu assay. A volume of 75 µL of diluted honey was mixed with 2925 µL of DPPH 25 mg/L solution in methanol and incubated for 30 min. Absorbance was measured at 515 nm. Results were expressed as mg equivalent Trolox per 100 mL of honey (mg eq./100 mL).

### 4.6. Honey Extracts for GC-MS

To extract the nonpolar compounds, 10 g of honey was weighed and mixed with 100 mL of methanol. After stirring, 100 mL of MilliQ water was added, followed by 50 mL of dichloromethane. The mixture was stirred again and left to rest for 20 min to separate the denser part with dichloromethane. The fraction was then allowed to dry in the fume hood. Finally, the dry extract was analyzed using GC-MS (adapted from Galgowski et al.) [81].

### 4.7. Gas Chromatography Analysis (GC-MS)

The phytochemical investigation of honey dichloromethane extract from *Euterpe oleracea* was conducted using a GC-MS Ultra system equipped with a single quadrupole mass spectrometer (GCMS-QP2010 Plus, Shimadzu, Canby, OR, USA) and a ZB-5HTS Inferno™ column. The column dimensions were 30.0 m length, 0.25 mm diameter, and 0.25 μm thickness. Experimental parameters of the GC-MS system were set as follows: column oven temperature: 60 °C, carrier gas: helium at a flow rate of 1.8 mL/min, injection temperature: 280 °C, injection mode: split ratio: 5.0, flow control mode: linear velocity of 48.9 cm/s, pressure: 16.2 psi, total flow: 13.8 mL/min, purge flow: 3.0 mL/min. The GC program included an ion-source temperature of 280 °C, interface temperature of 280 °C, solvent cut time of 3 min, detector gain of 0.97 kV ± 0.00 kV, and a threshold of 1000. Mass spectrometry analysis was performed in full-scan mode with a scan range of 37 to 660 *m*/*z*. Volatile organic compounds (VOCs) were identified by comparing their MS fragments with those in the Wiley Registry/NIST Mass Spectral Library. Each VOC was relatively quantified by normalizing against the total peak areas of VOCs. 

### 4.8. Statistical Analysis 

To determine whether there is a statistically significant difference between açaí floral nectar honey and wild honey, each sample was analyzed in triplicate. Analysis of variance (ANOVA) was chosen because it allows the means of more than two groups to be compared simultaneously in order to identify significant statistical differences between them. As the aim was to compare the means of the different types of honey, ANOVA was the appropriate tool to check whether these differences were significant, with a significance level of *p* < 0.05.

After ANOVA, Tukey’s test was applied to carry out multiple comparisons between the samples. Tukey’s test is particularly useful after ANOVA, as it controls for type I error and provides a means of identifying which specific groups differ from each other, at a 5% probability level. These analyses were conducted using Jamovi software (version 2.3) [82].

## 5. Conclusions

This study demonstrates that honeys predominantly composed of açaí pollen exhibit significantly higher levels of antioxidant compounds and superior antioxidant capacity compared to other types of honey. The identification of constituent compounds through GC-MS analysis further supports these findings. Moreover, these honeys contribute to the beneficial properties of honey, including antimicrobial, anti-inflammatory, and gastroprotective activities, thereby enhancing their nutritional and therapeutic value.

The study underscores the importance of considering the botanical origin of pollen when evaluating the antioxidant quality of honey. The presence of bioactive compounds in Brazilian honeys, particularly those from the Amazon region and derived from açaí nectar, highlights the unique potential of these products in promoting health and well-being. Combining these compounds with the recognized superfood properties of açaí opens new avenues for the appreciation and international marketing of Brazilian honey.

In conclusion, the distinct quality and nutritional richness of açaí honeys position them as valuable commodities in the global market, offering unique health benefits and contributing to the diversity of natural products available from Brazil.

## Figures and Tables

**Figure 1 molecules-29-04567-f001:**
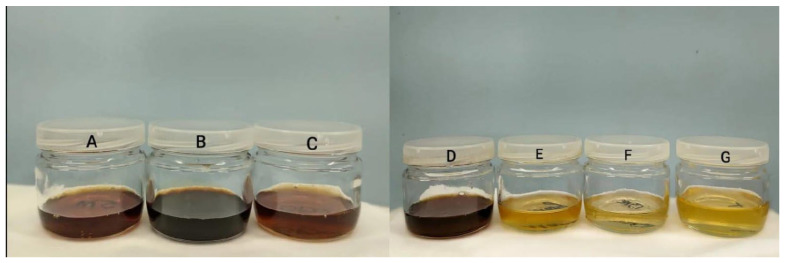
Honeys from different nectar floral of Brazil. Figure legend: Honeys of nectar floral açaí AH1 (**A**), nectar floral açaí AH2 (**B**), nectar floral açaí AH3 (**C**), and honey of Aroeira (**D**), honey of Mangue (**E**), honey of Cipó-uva (**F**), and honey of Timbó (**G**).

**Figure 2 molecules-29-04567-f002:**
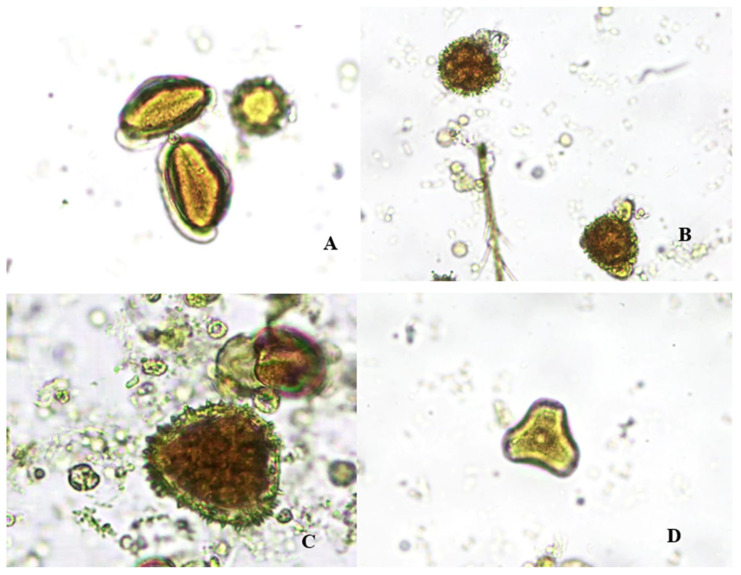
Pollens found in açaí honeys in AH1, AH2, and AH3. Figure legend: Photomicrographs of the melissopalynological analysis of pollen grains (×400), observed in AH1, AH2, AH3 (**A**) *Euterpe oleracea* pollen; (**B**) *Sida galheirensis* pollen; (**C**) *Acacia sp.* pollen; (**D**) *Euphorbiaceae* pollen.

**Figure 3 molecules-29-04567-f003:**
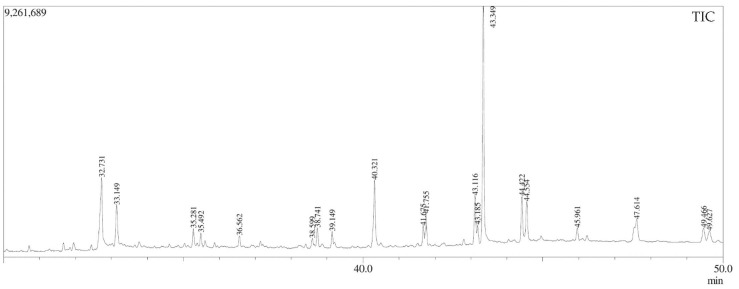
GC-MS Chromatogram of Açaí honey from the Breu Branco (AH1). Figure Legend: Sample of açaí honey with the highest percentage of area in the compounds of batilol, Oleic acid, Stearic acid, Heneicosane, Pentacosane, Heptacosanol, Tetratetracontane.

**Table 1 molecules-29-04567-t001:** The colors and intensity according to the Pfund scale.

Honeys	Color (mm Pfund)	Color Range	Intensity (mAU)
AH1	380.0	Dark amber	2.361
AH2	72.2	Amber	0.953
AH3	90.3	Amber	1.463
Aroeira	223.3	Dark amber	1.093
Mangue	61.7	Light amber	0.244
Cipó-Uva	42.2	Extra amber	0.111
Timbó	72.0 nm	Light amber	0.208

Table legend: Colors related to mm Pfund scale and optical density.

**Table 2 molecules-29-04567-t002:** Physicochemical parameters obtained from wild honeys and açaí flower nectar.

Honeys	Free Acidity (mEq/kg)	pH	Humidity (g/100 g)	°Brix (g/g)	Reducing Sugars (g/100 g)	Apparent Sucrose (g/100 g)
AH1	73.60 ± 2.9 ^a^	3.35 ± 0.0	19.82 ± 0.0	78.50 ± 0.0	62.76 ± 0.2	5.57 ± 0.3
AH2	63.03 ± 0.2 ^a^	3.54 ± 1.7	20.70 ± 0.0 ^b^	77.67 ± 0.0	64.26 ± 0.6	6.21 ± 0.3 ^b^
AH3	75.97 ± 0.8 ^a^	3.44 ± 0.0	19.5 ± 0.0	78.80 ± 0.1	64.49 ± 0.4	4.59 ± 0.0
Aroeira	30.74 ± 0.0	4.70 ± 0.1	16.50 ± 0.0	81.77 ± 0.2	62.26 ± 0.5	5.60 ± 0.1
Mangue	12.31 ± 0.2	4.20 ± 0.0	19.40 ± 0.0	79.00 ± 0.0	71.08 ± 0.3	0.74 ± 0.0
Cipó-Uva	14.62 ± 0.0	3.87 ± 0.0	16.50 ± 0.0	81.77 ± 0.0	67.75 ± 1.1	3.53 ± 0.1
Timbó	9.19 ± 0.2	4.23 ± 0.0	17.30 ± 0.0	81.03 ± 0.0	68.06 ± 0.4	1.38 ± 0.0

Table legend: Physio-chemical aspects of wildflower and açaí honeys expressed as mean and standard deviation; açaí honey from Breu Branco (AH1 and AH2) and Santa Maria (AH3); ^a^: *p* < 0.05 açaí honeys with wild honeys; ^b^: AH2 exceeds the limit allowed by Brazilian legislation (*p* < 0.05).

**Table 3 molecules-29-04567-t003:** The results of the spectrophotometric analyses obtained from açai floral nectar honey and other honeys.

Honeys	Polyphenol Content (mg eq./100 g)	Flavonoids Content (mg eq./100 g)	Flavanol Content (mg eq./100 g)	DPPH (µmol eq./100 mL)
AH1	291.84 ± 8.1 ^c^	106.05 ± 5.3 ^c^	1.95 ± 0.1	396.55 ± 10.4 ^c^
AH2	79.49 ± 2.4 ^d^	9.71 ± 0.3	1.21 ± 0.1	114.13 ± 3.2 ^d^
AH3	118.20 ± 5.2 ^c^	14.36 ± 0.9 ^e^	1.27 ± 0.1	157.81 ± 6.9 ^c^
Aroeira	100.05 ± 0.8	30.24 ± 0.9	4.20 ± 0.1	98.58 ± 1.2
Mangue	32.24 ± 0.8	7.43 ± 0.6	ND	15.05 ± 0.6
Cipó-Uva	25.15 ± 0.7	3.69 ± 0.1	ND	23.06 ± 0.9
Timbó	34.39 ± 2.3	9.67 ± 0.2	ND	14.07 ± 0.7

Table legend: Results of spectrophotometric analysis of açaí honeys (AH1, AH2, AH3) and Aroeira, Timbó, Mangue, and Cipó-Uva honeys presented as mean and standard deviation; ^c^: *p* < 0.001 in relation to wild honeys; ^d^: *p* < 0.001 with wild honeys, except Aroeira; ^e^: *p* < 0.05 only with Mangue and Cipó-Uva; ND, not detected (below the level of detection).

**Table 4 molecules-29-04567-t004:** Bioactive compounds present in samples of Açaí honey from the Breu Branco Region (AH1) by CG-MS.

Peak#	RT (min)	Fragments (%)	Name of Compound	Molecular Formula	MW (g/mol)	Ref.
1	8.82	57 (100); 41 (78.51); 43 (66.4)	Nonanal	C_9_H_18_O	142	-
2	12.20	97 (100); 41 (71.2); 126 (49.4)	5-Hydroxymethylfurfural	C_6_H_6_O_3_	126	[22]
3	18.81	55 (100); 73 (83.1); 41(78.1)	9-Oxononanoic acid	C_9_H_16_O_3_	172	-
4	29.45	73 (100); 43 (89.0); 60 (63.9)	Palmitic acid	C_16_H_32_O_2_	256	[23]
5	32.73	55 (100); 69 (83.9); 41 (83.5)	Oleic Acid	C_18_H_34_O_2_	282	[24]
6	33.15	55 (100); 43 (96.0); 73 (92.6)	Stearic acid	C_18_H_36_O_2_	284	-
7	35.28	83 (100); 69 (92.0); 57 (60.4)	Behenic alcohol	C_22_H_46_O	326	[25]
8	35.50	57 (100); 71 (79.9); 43 (46.7)	Heneicosane	C_21_H_44_	296	[26]
9	36.56	-	No identified	-	-	-
10	38.60	57 (100); 83 (88.4); 97 (58.7)	Heptacosanol	C_27_H_56_O	396	[27]
11	38.74	57 (100); 71 (82.5); 43 (23.7)	Eicosane	C_20_H_42_	282	-
12	39.15	149 (100); 167 (29.4); 57 (54.7)	Bis(2-ethylhexyl) phthalate	C_24_H_38_O_4_	390	-
13	40.32	57 (100); 71 (77.9); 43 (81.5)	Batilol	C_21_H_44_O_3_	344	[28]
14	41.67	83 (100); 97 (96.8); 69 (54.8)	Heptacosanol	C_27_H_56_O	396	[27]
15	41.75	57 (100); 71 (84.2); 85 (57.9)	Pentacosane	C_25_H_52_	352	[29]
16	43.12	185 (100); 57 (44.7); 70 (42.1	Decanedioic acid,bis(2-ethylhexyl)ester	C_26_H_50_O_4_	426	[30]
17	43.18	57 (100); 71 (83.7); 85 (63.7)	Tetracontane	C_40_H_82_	562	[31]
18	43.35	57 (100); 71 (62.6); 85 (49.4)	Batilol	C_21_H_44_O_3_	344	[28]
19	44.42	57 (100); 97 (89.8); 83 (82.7)	Heptacosanol	C_27_H_56_O	396	[27]
20	44.55	57 (100); 71 (81.9); 85 (57.9)	Tetratetracontane	C_44_H_90_	618	[32]
21	45.96	57 (100); 71 (83.7); 85 (63.7)	Tetracontane	C_40_H_82_	5632	[31]
22	47.61	57 (100); 71 (83.7); 85 (63.7)	Tetracontane	C_40_H_82_	562	[31]
23	49.47	55 (100); 314(94.2); 81(80.1)	Fucosterol	C_29_H_48_O	412	-
24	49.63	57 (100); 71 (85.0); 71 (84.2)	Campesterol	C_28_H_48_O	400	-
25	51.65	43 (100); 107 (98.4); 95 (87.3)	Sitosterol	C_29_H_50_O	414	[33]

Table legend: bioactive compounds present in samples of Açaí honey from the Breu Branco Region (AH1) detected by GC-MS.

**Table 5 molecules-29-04567-t005:** Bioactive compounds present in samples of honeys and açaí honeys.

Compound	Molecular Formula	Honey	Function	Ref.
Oleic Acid	C_18_H_34_O_2_	AH1, AH3 Timbó, Aroeira and Mangue	anti-inflammatory	[24]
5-Hydroxymethylfurfural	C_6_H_6_O_3_	AH1, AH3	quality indicator	[22]
n-Hexadecanoic acid	C_16_H_32_O_2_	AH1, AH2 Timbó, Aroeira and Mangue	Antioxidant and antibacterial	[23]
Batilol	C_21_H_44_O_3_	AH1, AH3 and Timbó	antitumor	[28]
n-Tetracosanol-1	C_24_H_50_O	-	-	-
Pentacosane	C_25_H_52_	AH1, Mangue, Timbó	antimicrobial	[29]
Heptacosanol	C_27_H_56_O	AH1, AH2 Mangue, Timbó and Aroeira	antioxidant	[27]
Octacosanol	C_28_H_58_O	AH3 and Aroeira	anti-inflammatory	[34]
9,12-Octadecadienoic acid, ethyl ester	C_20_H_36_O_2_	Mangue	antibacterial	[35]
Ethyl Oleate	C_20_H_38_O_2_	Mangue, AH3	Antioxidant	[36,37]
Heneicosane	C_21_H_44_	AH3, Mangue, Timbó and Aroeira	antimicrobial	[26]
2-Methylhexacosane	C_27_H_56_	AH2, AH3, Mangue and Timbó	antitumor	[38]
Behenic alcohol	C_22_H_46_O	AH1, AH3, Timbó, Mangue and Aroeira	antibacterial	[25]
Oxirane, [(dodecyloxy)methyl]-	C_15_H_30_O_2_	Mangue, Timbó and Aroeira	antifungal	[39]
Tetratetracontane	C_44_H_90_	Mangue, Timbó and Aroeira	antibacterial	[32]
Sitosterol	C_29_H_50_O	AH3, Mangue, Timbó,	antitumor	[33]
Hexadecanoic acid, ethyl ester	C_18_H_36_O_2_	Cipó-uva, AH3	antioxidant	[40]
Tetracontane	C_40_H_82_	AH1, Timbó, Cipó-uva, Aroeira	antimicrobial	[32]
Linoleic acid Ethyl ester	C_20_H_36_O_2_	Cipó-uva	antibacterial	[35]
beta-amyrin acetate	C_32_H_52_O_2_	AH2	anti-inflammatory, anticonvulsant and antibacterial	-
Octadecanoic acid, ethyl ester	C_20_H_38_O_2_	Cipó-uva	Antioxidant	[41]
Lupeol acetate	C_32_H_52_O_2_	AH2	Anti-inflammatory	-
1,2-Benzenedicarboxylic acid, bis(2-methyl)	C_16_H_18_O_4_	AH3, Timbó and Mangue	-	[42]
Methyl 9-octadecenoate	C_19_H_36_O_2_	AH2 and AH3	Antioxidant, Anti-inflammatory and antibacterial	-
Dotriacontane	C_32_H_66_	AH2	antioxidant	[43]
17-pentatriacontene	C_35_H_70_	Mangue	anti-inflammatory	-
Decanedioic acid, bis(2-ethylhexyl) ester	C_26_H_50_O_4_	AH1, AH3, Timbó and Mangue	antimicrobial	[30]
9-t-Butyltricyclo [4.2.1.1(2,5)]decane-9,10-diol	C_14_H_24_O_2_	Timbó	antibacterial	[44]
Spiro[bicyclo [3.1.1]heptane-2,2′-oxirane]	C_8_H_12_O	Mangue	anti-inflammatory, antimicrobial, and antioxidant	-
1-Hexacosanol	C_26_H_54_O	Timbó, Mangue, Aroeira,	antioxidant	-

Table legend: Bioactive compounds present in samples of honeys and açai honeys detected by GC-MS, as well as their molecular structure and action.

## Data Availability

Data are contained within the article and Appendix A.

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
