# Peer review of "Evaluation of the Bioactive Compounds of Apis mellifera Honey Obtained from the Açai (Euterpe oleracea) Floral Nectar"

_molecules, 2024, doi:10.3390/molecules29194567_

Round 1

Reviewer 1 Report

Comments and Suggestions for Authors

Keywords

Please include ““antioxidant, GC-MS/MS”” in the list of keywords.

Introduction

-          Lines 55-56: “This honey is obtained from the male and female flowers of the açaí palm tree (Euterpe oleracea), which produce nectar”, rephrase.

-          Give short note about used chromatography and spectroscopic methods for identifying  of honey’s metabolites

Results

-          Lines 78-88:” The results Açaí Honey from Breu Branco (AH1) were obtained from………. pollens with 2.67% (Figure 1)”, rephrase and avoid repeating.

-          Figures: captions and legends should be merging as figure caption below each figure.

Plant species should be italic.

-          Tables: Captions and legends should be merging as table caption below each table.

-          Line 140-141: “Although pH analysis is not mandatory for the quality control of………and durability”, rephrase

-          Figures 2-6 Captions: define AH1, AH3, AH2,……. Cipó-uv.

-          Table 5, 6: Change “C9H18O” to “C9H18O”, revise molecular formula to be in correct format.

-          Figure 7: Add assignment of major compounds.

-          Figure 7: Consider including GC-MS chromatography of other honey samples alongside AH1.

Materials and methods

-          Line 536 “100% methanol” and line 542: “honey ethanol extract”, what is the type of honey extract used for GC-MS analysis, clarify  

-A graphic abstract would greatly enhance the overall presentation of the paper. It is highly recommended.

Comments on the Quality of English Language

- The manuscript should be revised to correct the typo error

Author Response

Comments 1: Please include ““antioxidant, GC-MS/MS”” in the list of keywords.

Line 33-34. Response 1: Thank you for your insightful comment. We agree with your suggestion and have added "Antioxidant" and "GC-MS/MS" to the list of keywords, as requested.

Comments 2: Lines 55-56: “This honey is obtained from the male and female flowers of the açaí palm tree (Euterpe oleracea), which produce nectar”, rephrase.

Lines 59-61. Response 2: Thank you for your comment regarding the rephrasing of the sentence. We have reviewed and revised, and it now reads: "This honey is sourced from the nectar of both male and female flowers of the açaí palm tree (Euterpe oleracea)."

Comments 3: Give short note about used chromatography and spectroscopic methods for

identifying honey's metabolites.

Lines 77-79. Response 3: Thank you for your comment regarding the addition of methods to the text. In response, we have added the following: “using physicochemical parameters, quantification of phenolic compounds through spectroscopic methods, and chemical profiling by gas chromatography coupled with mass spectrometry (GC-MS).”

Comments 4: Lines 78-88:” The results Açaí Honey from Breu Branco (AH1) were obtained

from.......... pollens with 2.67% (Figure 1)”, rephrase and avoid repeating.

Lines 118-124. Response 4: Thank you for pointing out the repetitions in the text. We have revised the sentence to improve cohesiveness and eliminate repetitions. The revised version reads: “For all the honeys, a count of 500 pollen grains was conducted, identifying various pollen types. AH1 honey contains 50.5% Euterpe oleracea pollen, 20.48% Sida galheirensis pollen, 25.25% Acacia sp. pollen, and 3.78% Euphorbiaceae pollen. AH2 honey consists of 44.98% Sida galheirensis pollen, 20.8% Euterpe oleracea pollen, 17.8% Euphorbiaceae pollen, 3.42% Acacia sp. pollen, and 3% other pollen. AH3 honey contains 45.8% Euterpe oleracea pollen, 26.33% Sida galheirensis pollen, 17.33% Euphorbiaceae pollen, 7.87% Acacia sp. pollen, and 2.67% other pollen (Figure 2).”

Comments 5: Figures: captions and legends should be merging as figure caption below each

figure.

Response 5: Thank you for your comment regarding the separate captions and explanatory texts. We have reviewed and revised them in accordance with the journal's guidelines, as requested.

Comments 6: Tables: Captions and legends should be merging as table caption below each

table.

Response 6: Certainly. Thank you for your comment regarding the captions and explanatory texts for the tables. We have reviewed and adjusted them according to the journal's guidelines, consolidating them into a single caption below each table, as requested.

Comments 7: Line 140-141: “Although pH analysis is not mandatory for the quality control

of.........and durability”, rephrase

Lines 149-151. Response 7: Thank you for your comment. We have reviewed and revised the sentence for greater clarity and coherence, now rewritten as: “Although pH analysis is not mandatory for the quality control of Brazilian honey, legislation stipulates that the pH of Apis mellifera honey must be between 3.3 and 4.6 for human consumption.”

Comments 8: Figures 2-6 Captions: define AH1, AH3, AH2,....... Cipó-uv.

Response 8: Thank you for your comment. We have defined caption of figures 2-6 in as suggested, “ Honeys of nectar floral açaí AH1 (A), nectar floral açaí AH2 (B), nectar floral açaí AH3 (C) , and honey of Aroeira (D), honey of Mangue (E) Honey of Cipó-uva (F) and Honey of  Timbó (G).”

Comments 9: Table 5, 6: Change “C9H18O” to “C H O”, revise molecular formula to be in

correct format.

Response 9: Thank you for your comment regarding the molecular formula of the compounds. We have reviewed and corrected the format in Tables 4 and 5.

Comments 10: Figure 7: Add assignment of major compounds. Figure 7: Consider including GC-MS chromatography of other honey samples alongside AH1.

Lines 262-263. Response 10: Thank you for your comment and suggestion. We have added the major compounds identified by GC-MS in the AH1, we switch to Figure 3 with legend: “Sample of açaí honey with the highest percentage of area in the compounds of batilol, Oleic acid, Stearic acid, Heneicosane,Pentacosane, Heptacosanol, Tetratetracontane” However, we have decided to include only the GC-MS chromatogram of AH1 to specifically highlight the results from the açaí honey, particularly from the AH1 sample.

Comments 11: Line 536 “100% methanol” and line 542: “honey ethanol extract”, what is the type of honey extract used for GC-MS analysis, clarify

Line 555. Response 11: Thank you for pointing that out. We have reviewed the article and corrected the sentence ”dichloromethane extract”.

Comments 12: A graphic abstract would greatly enhance the overall presentation of the paper. It is highly recommended.

Response 12: Certainly. Thank you for your comment. The graphic abstract has been revised according to the recommendations.

4. Response to Comments on the Quality of English Language

Point 1: The manuscript should be revised to correct the typo error.

Response 1: Agreed. We have reviewed and corrected the typo errors.

Reviewer 2 Report

Comments and Suggestions for Authors

The authors have evaluated seven types of honey from different regions. They investigated the honey samples in terms of physicochemical properties, color, antioxidant content, and other significant constituents. They correlated the differences in properties and contents with flavor and aroma characteristics. The authors have used GC-MS as a method for identification and quantitative comparison of the unknown compounds in the honey extracts, however, there are some concerns that need to be addressed regarding the GC-MS data analysis which will be mentioned in detail. This paper could be publishable after major revisions.

- The authors have discussed the previous works on investigation of properties and contents of various honey samples. However, it is not clear what is the novelty of this work compared to previously published studies. I suggest that the authors emphasize the novelty of their research in the Introduction section.

- The retention times are not accurate to the third decimal place in chromatography. Please correct this issue in all figures and tables.

- In Tables related to GC-MS studies, the authors are reporting retention times vs. analytes. However, there are many cases in all the tables in the manuscript and the Supplementary Information section, that two or more retention times have been related to one analyte. For example, Table 4 has three 1-heptacosanols with different retention times. This is not possible.

The GC-MS libraries can be used to provide preliminary information regarding the content of an unknown mixture, but they cannot be trusted 100% as it can be observed that the same analyte was reported for multiple peaks with multiple retention times. I strongly suggest that the authors obtain the standards of the  key analytes that they compare in different honey samples and confirm the correct retention times for each analyte instead of relying fully on GC-MS libraries.

- Regarding the GC-MS studies, it should also be noted that a lot of the peaks were not separated with enough resolution. This makes the area% and area values reported in the tables and discussions in lines 355-438 inaccurate. Please improve the chromatographic method to ensure the baseline resolution of all peaks of the key analytes that are being compared in different honey samples.

- Have the authors run the GC-MS for each analyte in triplicate? There are chromatograms (Figures S2, S3, S5) that show column bleed or a carryover peak. This makes the quantitation inaccurate. Since the authors are reporting and comparing area values of various analytes, I would suggest that they run these chromatograms in triplicate to insure there are no carryovers. Additionally, if the same GC column and isocratic method was used in all chromatograms, it would not make sense if one chromatogram shows column bleed (Figure S2) and the others don’t.

- Could the authors please specify what solvent was used to do extraction from the honey samples? In line 536 methanol is mentioned as the extracting solvent and in line 542, ethanol is mentioned. Please clarify.

- Section 4.7: Please specify the injection volume.

- Please correct the chemical formulas used in the tables and through the whole manuscript, i.e., please use C27H56O instead of C27H56O.

- Please define the abbreviations when they are used for the first time. For example, DPPH and GC-MS.

- Please plot the chromatograms using Excel or MATLAB and avoid using screenshots.

- The label, “Table 4”, is missing for the table on page 11.

Author Response

 3. Point-by-point response to Comments and Suggestions for Authors

All added sentences of response to comments in the revised manuscript were highlighted in  orange color in the Reviewer’s Response.

Comments 1: The authors have discussed the previous works on investigation of properties and contents of various honey samples. However, it is not clear what is the novelty of this work compared to previously published studies. I suggest that the authors emphasize the novelty of their research in the Introduction section.

Response 1: Thank you for pointing this out. We emphasize the novelty of the article in the introduction as requested, we are grateful for this contribution to the improvement of the article.

Comments 2: The retention times are not accurate to the third decimal place in chromatography. Please correct this issue in all figures and tables.

Response 2:  Thank you for pointing this out. We corrected retention times to the correct decimal places in figures and tables.

Comments 3: In Tables related to GC-MS studies, the authors are reporting retention times vs. analytes. However, there are many cases in all the tables in the manuscript and the Supplementary Information section, that two or more retention times have been related to one analyte. For example, Table 4 has three 1-heptacosanols with different retention times. This is not possible. The GC-MS libraries can be used to provide preliminary information regarding the content of an unknown mixture, but they cannot be trusted 100% as it can be observed that the same analyte was reported for multiple peaks with multiple retention times. I strongly suggest that the authors obtain the standards of the  key analytes that they compare in different honey samples and confirm the correct retention times for each analyte instead of relying fully on GC-MS libraries.

Response 3: Thank you for pointing this out. The appearance of the same compound at different retention times is due to the presence of isomers, and because the absence of a chiral column, it was not used to differentiate these isomers. Therefore, some compounds are repeated throughout the run.

Comments 4: Regarding the GC-MS studies, it should also be noted that a lot of the peaks were not separated with enough resolution. This makes the area% and area values reported in the tables and discussions in lines 355-438 inaccurate. Please improve the chromatographic method to ensure the baseline resolution of all peaks of the key analytes that are being compared in different honey samples.

Response 4: The low resolution is due to the lack of derivatization, resulting in compounds with high polarity interacting with the silica column and not providing sufficient chromatographic resolution.

Comments 5: Have the authors run the GC-MS for each analyte in triplicate? There are chromatograms (Figures S2, S3, S5) that show column bleed or a carryover peak. This makes the quantitation inaccurate. Since the authors are reporting and comparing area values of various analytes, I would suggest that they run these chromatograms in triplicate to insure there are no carryovers. Additionally, if the same GC column and isocratic method was used in all chromatograms, it would not make sense if one chromatogram shows column bleed (Figure S2) and the others don’t.

Response 5:  In the case of the bleeding, due to the lack of derivatization, a higher temperature was likely required in the column oven, causing the sibilant compounds to interact with the analyte.

Comments 6: Could the authors please specify what solvent was used to do extraction from the honey samples? In line 536 methanol is mentioned as the extracting solvent and in line 542, ethanol is mentioned. Please clarify.

Line 550 and 555. Response 6: Thank you for your comment. We have corrected the extraction solvent.

Comments 7: Section 4.7: Please specify the injection volume.-

section 4.7. Response 7: Thank you for your comment. We have corrected the injection volume in “ injection mode: split ratio: 5.0

Comments 8: Please correct the chemical formulas used in the tables and through the whole manuscript, i.e., please use C H O instead of C27H56O. 27 56

Response 8:  Thank you for your comment. We have corrected the molecular formulas in all tables.

Comments 9: Please define the abbreviations when they are used for the first time. For example, DPPH and GC-MS.

Lines 79-280. Response 9: Agreed. We have reviewed and made the necessary corrections.

Comments 10: The label, “Table 4”, is missing for the table on page 11.

Line 265. Response 7: Agreed. We have reviewed and corrected it, and the update.

Comments 11: Please plot the chromatograms using Excel or MATLAB and avoid using screenshots.

Response 8: Thank you for your comment. We look and  improved the resolution of the chromatograms  in the article

Reviewer 3 Report

Comments and Suggestions for Authors

The manuscript entitled Evaluation of the Bioactive Compounds of Apis Mellifera Honey Obtained From The Açai (Euterpe oleracea) Floral Nectar presents a comprehensive evaluation of bioactive compounds in honey derived from the Açai floral nectar, a topic of significant interest due to the potential health benefits of such natural products. The study's focus on the comparison between Açai honey and other Brazilian honeys is novel and contributes to the existing literature on honey's phytochemical properties. The manuscript is well-written and structured, with clear sections that logically flow from one to the next.

Suggestions are as follows:

   - The manuscript could benefit from a more detailed explanation of the statistical tests used and the rationale behind selecting specific tests.

   - While the study is thorough, it may be beneficial to include a broader range of honey types in future research to further validate the findings.

   - The authors might consider discussing the practical applications of their findings, such as how Açai honey could be incorporated into dietary or medicinal practices.

- For the antioxidant assay, positive controls should be added.

- In the abstract, Apis mellifera L., L should not be in italic. 

Author Response

3. Point-by-point response to Comments and Suggestions for Authors

All added sentences of response to comments in the revised manuscript were highlighted in purple color in the Reviewer’s Response

Comments 1: The manuscript could benefit from a more detailed explanation of the statistical tests used and the rationale behind selecting specific tests.

Lines 571-581. Response 1: Thank you for your feedback. We agree with your suggestion and have revised the explanation of the statistical tests used, providing more detail on the rationale for selecting specific tests.

Comments 2: While the study is thorough, it may be beneficial to include a broader range of honey types in future research to further validate the findings.

Response 2: Thank You for your comment. We are grateful for your suggestion in the study and are considering including a wide range of honeys in the future.

Comments 3: The authors might consider discussing the practical applications of their findings, such as how Açai honey could be incorporated into dietary or medicinal practices.

Response 3: Agreed. We have reviewed the article and noted that the health benefits of açai honey consumption are discussed in both the results and conclusion sections. Additionally, we highlight that its potential for further research.

Comments 4: For the antioxidant assay, positive controls should be added.

Response 4: Thank you for your observation. We have reviewed the methodology, and since the analysis utilizes a Trolox concentration curve for quantification, a positive control was not included.

Comments 5: In the abstract, “Apis mellifera L.”, “L” should not be in italic.

Response 5: Thank you for pointing that out; we have reviewed and corrected it.

Reviewer 4 Report

Comments and Suggestions for Authors

The aim of the reviewed article was to characterize açaí honeys produced by African bees (Apis mellifera) from different locations and to compare the profile of bioactive compounds in nectar honeys from açaí flowers with nectar honeys from wild flowers from various regions of Brazil.

The theoretical part is well prepared. The results were described correctly, although the usual description of the results is limited to listing the numerical values ​​given in the tables. The discussion of the results is also not prepared in an interesting and professional manner. Reading the discussion of the results with literature data is very unattractive for the reader. The discussion is made up of a text divided into several clearly separated paragraphs. Each paragraph concerns a different parameter or feature of the honeys analyzed. There is a lack of a "neat" connection of the individual paragraphs so that the whole creates an interesting story about the honeys studied.

 The methodological part requires improvement in accordance with the reviewer's instructions.

Below are my detailed comments.

1. I would consider swapping the order of Figures 1 and 2. It might be worth showing first what the test samples looked like (individual honey samples - Figure 2), and only then the figures related to the analysis of the composition of these samples (Figure 1). I understand the placement of Figure 2 in the place where the results of the analysis of the color of honeys are discussed, however in the text, when the results of the color of honeys are discussed, one can refer to the appropriate figure. Please consider this suggestion.

2. Table 2: Some results for the same feature (pH, humidity, brix) are given to one decimal place, and some to two decimal places. Please standardize.

3. Table 2 and 3: Why are the results of the statistical analysis not presented in the table?

4. Table 1, 2 and 3: Why is the order of honey samples not the same in the tables?

5. Figures 3, 4, 5 and 6 show the data presented in Table 3. It is assumed that the same results should not be presented twice. Please decide on either a tabular summary of the results or their presentation in the form of a figure.

6. In the case of the honey samples studied, was HPLC analysis used for qualitative and quantitative determination of polyphenols? HPLC analysis would provide a lot of interesting information on the polyphenol composition of honeys. Spectrophotometric methods are burdened with a certain error resulting from the fact that certain substances that are not polyphenolic compounds react with the Folin-Ciocalteu reagent. HPLC analysis would be recommended to assess the content of polyphenols, including flavonoids, in the honey samples tested.

Line 154: There should be a comma between the number and the unit.

Lines 157-159: It's a good idea to include the unit in this sentence.

Line 185 and table 3: It is worth specifying what the spectrophotometric analysis concerns (polyphenol, flavonoid content, etc.).

Lines 465-466: In the section on the discussion of results, the authors state that pH and Brix determinations are not required by Brazilian legislation. The content of the quoted lines indicates otherwise.

Line 534: There should be „GC-MS”.

Line 542: Is this an ethanol or methanol extract? In line 536 it says that methanol was added to the honey samples.

Author Response

3. Point-by-point response to Comments and Suggestions for Authors

All added sentences of response to comments in the revised manuscript were highlighted in green color in the Reviewer’s Response.

Comments 1: The theoretical part is well prepared. The results were described correctly, although the usual description of the results is limited to listing the numerical values given in the tables. The discussion of the results is also not prepared in an interesting and professional manner. Reading the discussion of the results with literature data is very unattractive for the reader. The discussion is made up of a text divided into several clearly separated paragraphs. Each paragraph concerns a different parameter or feature of the honeys analyzed. There is a lack of a "neat" connection of the individual paragraphs so that the whole creates an interesting story about the honeys studied.

Response 1: Thank you for pointing this out. We analyzed the points in the article discussion and evaluated the modifications to structure it more fluidly between paragraphs. Our discussion is based on clearly presenting the results found in each parameter analyzed in the honeys presented. Thank you for your contribution to improving the discussion.

Comments 2: I would consider swapping the order of Figures 1 and 2. It might be worth showing first what the test samples looked like (individual honey samples - Figure 2), and only then the figures related to the analysis of the composition of these samples (Figure 1). I understand the placement of Figure 2 in the place where the results of the analysis of the color of honeys are discussed, however in the text, when the results of the color of honeys are discussed, one can refer to the appropriate figure. Please consider this suggestion.

Lines 109 and 127. Response 2: Thank you for your suggestion. Upon review, we agree with the proposed adjustment and have reordered the figures as recommended.

Comments 3: Table 2: Some results for the same feature (pH, humidity, brix) are given to one decimal place, and some to two decimal places. Please standardize.

Lines 136 and 202. Response 3: We agree with your observation. We have revised the table values and standardized them in line with Table 3.

Comments 4: Table 2 and 3: Why are the results of the statistical analysis not presented in the table?

Lines 136 and 202. Response 4: Thank you for your observation. We have reviewed the manuscript, and the statistical analysis results are indicated in the tables using “ a “, “ b “ e “ c “ “d “ and “ e “ with an explanation provided in the table legend.

Comments 5: Table 1, 2 and 3: Why is the order of honey samples not the same in the tables?

Lines 88-93, 136 and 202. Response 5: Thank you for your feedback. We have reviewed and standardized the order of the honey samples across Tables 1, 2, and 3.

Comments 6: Figures 3, 4, 5 and 6 show the data presented in Table 3. It is assumed that the same results should not be presented twice. Please decide on either a tabular summary of the results or their presentation in the form of a figure.

Line 202. Response 6: We agree with your suggestion. Therefore, we have opted to present the data only in table format.

Comments 7: In the case of the honey samples studied, was HPLC analysis used for qualitative and quantitative determination of polyphenols? HPLC analysis would provide a lot of interesting information on the polyphenol composition of honeys. Spectrophotometric methods are burdened with a certain error resulting from the fact that certain substances that are not polyphenolic compounds react with the Folin-Ciocalteu reagent. HPLC analysis would be recommended to assess the content of polyphenols, including flavonoids, in the honey samples tested.

Response 7: Thank you for your feedback. Due to the presence of high levels of aromatic compounds in honey and their ability to withstand high temperatures, the choice of GC/MS over LC/MS was also influenced by the scarcity of data related to GC/MS. However, in future work, we plan to conduct a comparative study between GC/MS and LC/MS.

Comments 8: Line 154: There should be a comma between the number and the unit.

Lines 157-159: It's a good idea to include the unit in this sentence. Line 185 and table 3: It is worth specifying what the spectrophotometric analysis concerns (polyphenol, flavonoid content, etc.). Line 534: There should be „GC-MS”. Line 542: Is this an ethanol or methanol extract? In line 536 it says that methanol was added to the honey samples.

Lines 161-162, 197-198 and 554-555. Response 8: Agreed. We have reviewed the manuscript and made the necessary corrections.

Comments 9:  Lines 465-466: In the section on the discussion of results, the authors state that pH and Brix determinations are not required by Brazilian legislation. The content of the quoted lines indicates otherwise.

Lines 149-151, 344-345. Response 9: Certainly. Thank you for your comment. We reviewed and confirmed that, although Brazilian legislation does not require pH and Brix analysis for honey quality control, it does recommend that honey's pH should remain within the range of 3.3 to 4.6 for human consumption. Therefore, we have revised the sentences in lines 149-151 in the results section and lines 344-345 in the discussion section to make the text more cohesive and precise.

Round 2

Reviewer 2 Report

Comments and Suggestions for Authors

The authors have addressed most of the comments adequately. However, please see my comments on your responses below:

Response 3: Thank you for pointing this out. The appearance of the same compound at different retention times is due to the presence of isomers, and because the absence of a chiral column, it was not used to differentiate these isomers. Therefore, some compounds are repeated throughout the run.

Thank you for the explanation. Since a chiral column was not used, it is not possible that you could have separated enantiomers. Therefore, these could not have been enantiomers. Regarding other structural isomers, I believe that whatever you are reporting as constituents of honey should be tested using a standard. Otherwise, it is unreliable. I suggest you look at the similarity% that the MS library provides as well. If the number is not high enough, there is a high chance that it is incorrect considering all of the peak overlaps in the chromatograms. 

Response 4: The low resolution is due to the lack of derivatization, resulting in compounds with high polarity interacting with the silica column and not providing sufficient chromatographic resolution.

Thank you for the response. I understand, therefore you cannot report area% and do quantitative comparison accurately. 

Author Response

Point-by-point response to Comments and Suggestions for Authors

Comments 1: Thank you for the explanation. Since a chiral column was not used, it is not possible that you could have separated enantiomers. Therefore, these could not have been enantiomers. Regarding other structural isomers, I believe that whatever you are reporting as constituents of honey should be tested using a standard. Otherwise, it is unreliable. I suggest you look at the similarity% that the MS library provides as well. If the number is not high enough, there is a high chance that it is incorrect considering all of the peak overlaps in the chromatograms.

Response 1: Thank you for raising this point. We understand your position regarding the results in the table. However, given the numerous constituents found in honey, it is challenging to test every peak in the chromatogram using standards. As suggested, we have removed the % of area. Since the spectra were processed manually, we did not consider the % from the MS library provided.

Comments 2: Thank you for the response. I understand, therefore you cannot report area% and do quantitative comparison accurately.

Response 2: Thank you very much for your comment. We understand your point regarding the percentage of compounds in our table, and as a result, we have removed it.

Reviewer 4 Report

Comments and Suggestions for Authors

I have checked the corrections made by the authors. All suggestions have been taken into account. The article is suitable for publication.

Author Response

We sincerely appreciate the insightful comments and suggestions provided by the reviewers and you. These valuable inputs have allowed us to expand our research, refine our data, and enhance the quality of the manuscript. We are now pleased to submit the revised version for your consideration